# EPINN: Physics-Informed Neural Network with exponential activation functions for solving stiff ODEs

## Abstract

Solving stiff ordinary differential equations (ODEs) through machine learning methods has been quite a popular topic for years as it challenges the recently proposed physics-informed neural network (PINN). Many variations based on PINN have been advanced to enhance both the efficiency and the robustness. Nonetheless, many of them need to find the trade-off between the precision and speed because they have to train hundreds or even thousands of parameters if they do not design good or problem-adapt networks. In this scenario, we put forward a single layer physics-informed neural network with exponential activation functions (EPINN) by implementing the prior knowledge of the solution to the linear stiff ODEs. Under this simple but useful structure, less parameters would be sufficient and the model is easy to train. The model is also extended to solve nonlinear systems by introducing sequential EPINN. The network is tested on six benchmark problems including both linear and nonlinear ones and shows great performance.

## 1 Introduction

Stiff ordinary differential equations (ODEs) are frequently employed to describe variable phenomena of fields such as atmospheric chemistry, physics and chemical engineering. Formally, stiff refers to the fact that both fast and slow processes exist simultaneously, in which the absolute changes of different components sometimes could cover several orders of magnitude within a relatively narrow time interval, mathematically called boundary or inner layers. On the one hand, this property makes the numerical solving significantly challenging for both the traditional and machine learning schemes since one has to use very small time steps or a great number of neurons to capture the huge change of the solution. On the other hand, stiff ODE systems like Van Der Pol's equation and Rober equations are very essential models. As a result, finding highly efficient solver for stiff ODEs is a demanding and formidable work.

Generally speaking, ODEs can be integrated either by traditional schemes or machine learning methods. The traditional algorithms, such as finite difference method (FDM), mainly consist of explicit and implicit methods. Nevertheless, solving stiff ODEs is still a tough task because both efficiency and high stability are significantly in need especially for stiff ODEs. The explicit methods often do not satisfy the stability requirement while the implicit ones are computationally intensive most of the time.

With the development of machine learning, neural network methods have enabled advances in solving differential equations, among which the recently proposed physics-informed neural network (PINN) embeds physics laws in the training of the network, transferring data-driven methods into knowledge-driven methods Raissi et al. (2019). After being brought up, PINN has been successfully applied to plenty of issues from numerous fields such as solid mechanics Haghighat et al. (2021); Vahab et al. (2022); Niaki et al. (2021); Samaniego et al. (2020); Arora et al. (2022) and fluid mechanics Mao et al. (2020); Almajid & Abu-Al-Saud (2022).

Despite the great performance deep learning methods have shown in many areas, when facing multi-task or multi-scale problems, regular deep learning schemes are usually not so powerful. For instance, it is known that PINN have troubles in solving stiff ODEs Karniadakis et al. (2021). Reasons are analyzed Wang et al. (2021) and variations are raised such as stiff PINN, MPINN, vanilla PINN

and reduced PINN Ji et al. (2021); Weng & Zhou (2022); Baty (2023); Nasiri & Dargazany (2022). Although these models improve the performance of the original network, they all have to train hundreds or even thousands of parameters which is time-consuming. PIRPNN observes this issue and introduces a single layer network to reduce the number of parameters and enhance the efficiency by using the technique of random projection, while it still needs to lots of neurons to guarantee the accuracy Fabiani et al. (2023).

In this scenario, our interest is to construct a model with parameters as few as possible while maintaining both the accuracy and efficiency. Our main contributions can be summarized as follows:

- We put forward a single neural network with exponential activation functions by implementing prior knowledge of the solution to obtain a fast and accurate stiff ODE solver.
- Under our structure, very few parameter would suffice to get a satisfactory solution and the loss function is smooth and convex so that the global optimizer can be guaranteed.
- The model is tested on several benchmark stiff cases containing both linear and nonlinear ones and shows great performance.

The paper is organized as follows. In section 2, we introduce the basic information about our problem settings and preliminaries. In section 3, we discuss the method and construct our model. In section 4, we compare its performance with the famous stiff ODE solver ode15s in Matlab on six benchmark cases. Finally, we draw our conclusions in section 5.

## 2 PROBLEM SETTINGS AND PRELIMINARIES

### 2.1 PROBLEM SETTINGS

The initial value problem of the following stiff ODEs is of our interest:

$$\begin{cases} \boldsymbol{y}' = \boldsymbol{f}(t, \boldsymbol{y}), & t \in (0, T], \\ \boldsymbol{y}(0) = \boldsymbol{y}_0. \end{cases} \tag{1}$$

where $\boldsymbol{y}, \boldsymbol{f} \in \mathbb{R}^d$. Without generality, we can always assume the time start of equation 1 be 0 because we can always make a time translation $t \mapsto t - t_0$ if the equation starts at $t_0 \neq 0$.

### 2.2 CLASSICAL PHYSICS-INFORMED NEURAL NETWORK

With the pure data-driven deep learning methods achieving great break-through in many artificial intelligence domains, PINN is believed to transform the data-driven methods into knowledge-driven methods by embedding physics laws or prior knowledge in the design of the network. By implementing PINN, the training data could be reduced to several points, which is especially suitable for differential equations where only the initial and (or) boundary conditions are known in most cases.

In detail, a neural network can be represented by $\mathcal{N}(\boldsymbol{x})$ with $\boldsymbol{x}$ as input, and $\mathcal{N}$ as the output given by the following composite function 2:

$$\begin{aligned} \mathcal{N}(\boldsymbol{x}, \boldsymbol{\theta}) &= \boldsymbol{l} \circ \Phi^L \circ \cdots \circ \Phi^0(\boldsymbol{x}), \\ \Phi^i(\boldsymbol{x}) &= \sigma_i(\boldsymbol{W}^i \boldsymbol{x} + \boldsymbol{b}^i), i = 0, \ldots, L. \end{aligned} \tag{2}$$

where $\boldsymbol{\theta}$ denotes the set of weight matrices and bias vectors $\{\boldsymbol{W}^i, \boldsymbol{b}^i\}_{i=0}^L$, called adjustable parameters, $\sigma_i$ a nonlinear activation function, and $\boldsymbol{l}$ a linear amplification function.

Taking the following 1-dimensional partial differential equation (PDE) 3 with Dirichlet boundary condition as an example,

$$\begin{cases} \mathcal{L}u = 0, x \in \Omega, \\ u|_{x \in \partial \Omega} = h. \end{cases} \tag{3}$$

where $\mathcal{L}$ is a differential operator. PINN can be constructed by minimizing the following loss function:

$$L(\boldsymbol{\theta}) = \frac{\omega_b}{|\Omega_b|} \sum_{x \in \Omega_b} |\mathcal{N}(x, \boldsymbol{\theta}) - h(x)| + \frac{\omega_e}{|\Omega_e|} \sum_{x \in \Omega_e} |\mathcal{L}\mathcal{N}(x, \boldsymbol{\theta})|. \tag{4}$$

where $\Omega_b$ and $\Omega_e$ are two limited sample sets of $\partial\Omega$ and $int(\Omega)$, respectively. And $\omega_b, \omega_e$ are two positive adjustable parameters which can be pre-determined or self-adaptive. Finally, denoting the optimizer of 4 as $\boldsymbol{\theta}^*$, PINN regards $\mathcal{N}(x, \boldsymbol{\theta}^*)$ as the solution to equation 3.

According to the universal approximation theorem, neural networks offer a good approximation of a continuous function with sufficient layers and neuronsHornik et al. (1989). However, there is a dilemma between the accuracy and efficiency because multi-layer structure and too many neurons would make the model difficult to train although a great deal of techniques are employed to settle this problem.

## 2.3 RELATED WORKS

After being proposed, PINN has been successfully utilized to solve many classical differential equations including the Burgers' equation, the Navier-Stokes equation and the Schrodinger equation Raissi et al. (2019); Jin et al. (2021); Naderibeni et al. (2024). However, it has troubles in solving stiff ODEs. Afterwards, reasons are analyzed and an adaptive algorithm is put forward to balance the huge gradient difference among different loss terms Wang et al. (2021). Several variations based on PINN are brought up as well.

Different models emphasize on different aspects. Some transfer the original equations to make the training easier: Stiff PINN introduces quasi-steady state assumption to make the equations less stiffer Ji et al. (2021). As a result, the model learns the solution to the milder system rather than the original one. Reduced PINN transfers the governing equation into an integral one Nasiri & Dargazany (2022). However, the numerical solving of the integral has to be implemented in a relatively small time interval, making the method time-consuming. Some adjust the structure of the network: MPINN uses different networks to learn the fast and slow components separately but needs some ground truth data to guarantee the accuracy Weng & Zhou (2022), which is inaccessible in many cases. Vanilla PINN imposes some strategies like hybrid loss and moving collocation grid but cannot solve the strongly stiff cases Baty (2023). Some focus on the activation functions chosen: PIRPNN uses a single layer network with Gaussian kernels as its activation functions to approximate the solution Fabiani et al. (2023). It introduces random projection technique to reduce the parameters and shows great performance on nonlinear stiff ODEs.

## 3 METHOD

To guarantee the efficiency of the training, we also choose a single layer network to approximate the solution, which is to find a group of basis functions $\sigma(t) \in \mathbb{R}^d$ such that the solution $\boldsymbol{y}(t)$ to equation 1 can be approximated by:

$$\boldsymbol{y}(t) \approx \boldsymbol{y}_N(t) = \boldsymbol{W}\sigma(t). \tag{5}$$

where $\boldsymbol{W} \in \mathbb{R}^{d \times d}$. Different from PIRPNN, we use prior knowledge rather than random projection techniques such that less neurons and weight parameters would be sufficient. To achieve this goal, we first analyze the structure of the solution.

### 3.1 THE STRUCTURE OF THE SOLUTION

As nonlinear equation 1 can be approximated by the following linear system:

$$\begin{cases} \boldsymbol{y}' = \dfrac{\partial}{\partial \boldsymbol{y}}\boldsymbol{f}(t, \boldsymbol{y}_0)(\boldsymbol{y} - \boldsymbol{y}_0) + \boldsymbol{f}(t, \boldsymbol{y}_0), t \in (0, T], \\ \boldsymbol{y}(0) = \boldsymbol{y}_0. \end{cases} \tag{6}$$

Therefore, we only need to discuss the structure of linear ODEs.

### 3.1.1 LINEAR ODES WITH CONSTANT COEFFICIENTS

We first consider the following linear ODEs with constant coefficients:

$$\begin{cases} \boldsymbol{y}' = \boldsymbol{A}\boldsymbol{y} + \boldsymbol{b}, t \in (0, T], \\ \boldsymbol{y}(0) = \boldsymbol{y}_0. \end{cases} \tag{7}$$

where $\boldsymbol{A} \in \mathbb{R}^{d \times d}$ and $\boldsymbol{y}, \boldsymbol{b} \in \mathbb{R}^d$. Suppose $\lambda_1, \ldots, \lambda_m$ are m eigenvalues of $\boldsymbol{A}$ with geometric multiplicity $d_1, \ldots, d_m$, respectively and $|\lambda_1| \geq |\lambda_2| \geq \cdots \geq |\lambda_m| > 0$. The system 7 is called stiff if:

- $\Re(\lambda_i) < 0, i = 1, \ldots, d$,
- $\frac{|\lambda_1|}{|\lambda_m|} \gg 1$.

and the above ratio $\frac{|\lambda_1|}{|\lambda_m|}$ is called stiffness ratio or rigidity ratio. We can further assume the Jordan normal form of $\boldsymbol{A}$ is:

$$\boldsymbol{J} = \boldsymbol{P}^{-1} \begin{pmatrix} \boldsymbol{J}_1 & 0 & \cdots & 0 \\ 0 & \boldsymbol{J}_2 & \cdots & 0 \\ \vdots & \vdots & \ddots & \vdots \\ 0 & 0 & \cdots & \boldsymbol{J}_m \end{pmatrix} \boldsymbol{P}. \tag{8}$$

where $\boldsymbol{J}_i$ is the following $d_i \times d_i$ dimensional Jordan block matrix:

$$\boldsymbol{J}_i = \begin{pmatrix} \lambda_i & 1 & 0 & \cdots & 0 & 0 \\ 0 & \lambda_i & 1 & \cdots & 0 & 0 \\ 0 & 0 & \lambda_i & \cdots & 0 & 0 \\ \vdots & \vdots & \vdots & \ddots & \vdots & \vdots \\ 0 & 0 & 0 & \cdots & \lambda_i & 1 \\ 0 & 0 & 0 & \cdots & 0 & \lambda_i \end{pmatrix}. \tag{9}$$

Denoting $\sigma_i(t)$ as the column vector composed of:

$$\sigma_i(t) = (e^{\lambda_i t}; t e^{\lambda_i t}; \cdots ; t^{d_i - 1} e^{\lambda_i t}), i = 1, \ldots, m. \tag{10}$$

and $\sigma_{\boldsymbol{A}}(t)$ as $(\sigma_1(t); \ldots ; \sigma_m(t))$, it can be proved that the solution to equation 7 can be written as the linear combination of $(1; \sigma_{\boldsymbol{A}}(t))$ (seeing details in the appendix A.1).

### 3.1.2 LINEAR ODES WITH VARIABLE COEFFICIENTS

For linear system with variable coefficients:

$$\begin{cases} \boldsymbol{y}' = \boldsymbol{A}(t)\boldsymbol{y} + \boldsymbol{b}(t), t \in (0, T], \\ \boldsymbol{y}(0) = \boldsymbol{y}_0. \end{cases} \tag{11}$$

where $\boldsymbol{A}(t) \in \mathbb{R}^{d \times d}$ and $\boldsymbol{y}(t), \boldsymbol{b}(t) \in \mathbb{R}^d$. By introducing the technique in the appendix A.2, we can always assume equation 11 is a homogeneous system ($\boldsymbol{b}(t) \equiv 0$).

Suppose $\boldsymbol{A}(t) = \sum_{i=1}^{k} \frac{1}{i+1} \boldsymbol{A}_i t^i$ is a k-th order polynomial matrix function, the analytic solution to equation 11 can be written as the following form:

$$\boldsymbol{y}(t) = e^{\sum_{i=0}^{k} \boldsymbol{A}_i t^{i+1}} \boldsymbol{y}_0. \tag{12}$$

However, unlike the first simple case, the exponential matrix function 12 does not have explicit expression. Therefore, some asymptotic analysis techniques have to be introduced. By utilizing the skill of operator splittingSportisse (2000), $\Pi_{i=0}^{k} e^{\boldsymbol{A}_i t^{i+1}} \boldsymbol{y}_0$, denoted as $\bar{\boldsymbol{y}}(t)$, can be used to approximate $\boldsymbol{y}(t)$. The benefit is that $\bar{\boldsymbol{y}}(t)$ has explicit form and can be written as the linear combination of functions:

$$\sigma(t) = \otimes_{i=0}^{k} \sigma_{\boldsymbol{A}_i}(t). \tag{13}$$

where $\otimes$ is the Kronecker product and $\sigma_{\boldsymbol{A}_i}(t)$ is defined in the following way similar to 10:

$$e^{\lambda_j^i t^{i+1}}, t^{i+1} e^{\lambda_j^i t^{i+1}}, \cdots, (t^{i+1})^{d_j^i - 1} e^{\lambda_j^i t^{i+1}}, j = 1, \ldots, m_i, i = 0, \ldots, k. \tag{14}$$

where $\lambda_j^i$ is the eigenvalues of $A_i$ with multiplicity $d_j^i$ (seeing details in the appendix A.3).

For the case where $\boldsymbol{A}(t)$ is not a polynomial function, we hope to transform the equation into the one with polynomial coefficients, which is fully discussed. A natural intuition is to use the Taylor

expansion of $\boldsymbol{A}(t)$ to approximate $\boldsymbol{A}(t)$, which offers a polynomial function approximation. In detail, one can find the k-th order Taylor series of $\boldsymbol{A}(t)$ around the initial point as:

$$\boldsymbol{A}(t) = \sum_{i=0}^{k} \frac{\boldsymbol{A}^{(i)}(0)}{i!} t^i. \tag{15}$$

then it goes back to the polynomial case.

### 3.2 EXPONENTIAL ACTIVATION FUNCTIONS AND EPINN

As we already know that the solution to system 11 can be written as the linear combination of 13, we can implement 13 as the activation functions, which we denote as $\sigma(t)$ and call exponential activation functions. Afterwards, we apply the linear combination of $\sigma(t)$ as the output, denoted as $\mathcal{N}(t, \boldsymbol{W}) = \boldsymbol{W}\sigma(t)$, which we call exponential physics-informed neural network(EPINN). Under the format of PINN, the loss function can be constructed as:

$$L(\boldsymbol{W}) = \frac{\omega_i}{2}\|\mathcal{N}(0, \boldsymbol{W}) - \boldsymbol{y}_0\|_2^2 + \frac{\omega_e}{2n}\sum_{i=1}^{n}\|\frac{\partial}{\partial t}\mathcal{N}(t_i, \boldsymbol{W}) - \boldsymbol{A}(t_i)\mathcal{N}(t_i, \boldsymbol{W}) - \boldsymbol{b}(t_i)\|_2^2. \tag{16}$$

where $\Omega_e = \{t_1, \cdots, t_n\} \subset (0, T]$ is the sample set and $\omega_i, \omega_e$ are two positive adjustable parameters. Through minimizing the loss function $L(\boldsymbol{W})$, the optimizer of the training is regarded as the solution. It benefits a lot from constructing EPINN:

- **Accuracy of the solution:** By the discussion in section 3.1.1, we know our model can compute the exact solution for linear systems with constant coefficients. Additionally, for linear systems with variable coefficients and nonlinear systems, EPINN can also offer a good approximation.

- **Efficiency of the training:** We construct a single layer neural network similar to PIRPNN Fabiani et al. (2023), where only a few weight matrix parameters are needed to be optimized. Furthermore, less neurons would suffice as we implement the prior knowledge of the solution in the design of the network, which makes the training very fast.

- **Flexibility of the coding:** For any sample set, $L(\boldsymbol{W})$ is a smooth and convex function. Therefore, quite a few optimization algorithms such as the least square method and gradient descent algorithms can be used and the global minimum is guaranteed. Moreover, the gradient is easy to compute and there is no need to introduce auto differentiation or other alternatives, which would sharply reduce the complexity of the code.

The discussion above is based on the fact that we are informed with all the eigenvalues of the corresponding matrix. When the information about the eigenvalues are not or partly known, the network can also be trained by parameterizing all the eigenvalues of $\boldsymbol{A}$ with algebraic multiplicity d. In this case, both the weight matrix and the eigenvalue parameters are needed to be optimized. The loss function would be non-convex but still smooth. It is noteworthy that the more information about eigenvalues we know, the easier the training would be. Particularly, the model is quite useful for $d \leq 3$ as $\lambda_1$ and $\lambda_m$ can be quickly calculated by using power and inverse power algorithms under the stiff assumptionGolub & Van Loan (2013). And it is easy to compute all the eigenvalues by using the fact in matrix computation: $\sum_{i=1}^{m} d_i\lambda_i = \text{Tr}(\boldsymbol{A})$.

### 3.3 TECHNIQUES USED IN THE TRAINING

Several techniques are used to enhance the robustness of the model.

#### 3.3.1 SEQUENTIAL EPINN

The approximation of the nonlinear system 1, the approximation of the linear system with variable coefficients 11 and the use of operator splitting method 12 are all local properties, which means the computation can maintain high accuracy only in the small neighborhood of the initial point. Therefore, sequential EPINN is in need to improve the accuracy and robustness of the model.

We partition the integral span $[0, T]$ into $N$ sub-intervals: $[t_0, t_1] \cup [t_1, t_2] \cup \cdots \cup [t_{N-1}, t_N]$ where $t_0 = 0, t_N = T$. For $i = 1, 2, \ldots, N$, we solve the following sub-problem:

$$\begin{cases} \boldsymbol{y}_i' = \boldsymbol{f}(t, \boldsymbol{y}_i), t \in (t_{i-1}, t_i], \\ \boldsymbol{y}_i(t_{i-1}) = \boldsymbol{y}_{i-1}(t_{i-1}). \end{cases} \tag{17}$$

where $\boldsymbol{y}_0(t_0) = \boldsymbol{y}_0$. The partition can be pre-determined or self-adaptive. One can just use equidistant partition or introduce the techniques such as Fabiani et al. (2023); Huang & Seinfeld (2022) to obtain a self-adaptive time-step scheme.

### 3.3.2 SOLVING OF EACH OPTIMIZATION PROBLEM

For each sub problem, we need to solve the optimization problem in the following form:

$$\min_{\boldsymbol{W}} \quad \frac{\omega_i}{2} \|\boldsymbol{W}\sigma(0) - \boldsymbol{y}_0\|_2^2 + \frac{\omega_e}{2n} \sum_{i=1}^{n} \|\boldsymbol{W}\sigma'(t_i) - \boldsymbol{A}(t_i)\boldsymbol{W}\sigma(t_i) - \boldsymbol{b}(t_i)\|_2^2. \tag{18}$$

As we have proved that the above function 18 is convex for any $\boldsymbol{W} = (\boldsymbol{w}_1, \boldsymbol{w}_2, \ldots, \boldsymbol{w}_d) \in \mathbb{R}^{d \times d}$, the minimum can be obtained by solving the first order optimality condition. By writing $\boldsymbol{W}$ as $\bar{\boldsymbol{W}} = (\boldsymbol{w}_1; \boldsymbol{w}_2; \ldots; \boldsymbol{w}_d) \in \mathbb{R}^{d^2}$, one can prove the first order optimality condition is the following linear system(seeing details in the appendix A.4):

$$\boldsymbol{P}\bar{\boldsymbol{W}} = \boldsymbol{p}. \tag{19}$$

where $\boldsymbol{P} \in \mathbb{R}^{d^2 \times d^2}, \boldsymbol{p} \in \mathbb{R}^{d^2}$. And the solution is unique if and only if $\boldsymbol{P}$ is non-singular. In summary, the whole procedures of the algorithm can be written as:

---
**Algorithm 1** Algorithm of EPINN
---
**Input:** information about the ODEs
**Output:** solution on desired points
1: partition the interval $[0, T]$ into $[t_0, t_1] \cup [t_1, t_2] \cup \cdots \cup [t_{N-1}, t_N]$
2: **for** each $i = 1, 2, \ldots, N$ **do**
3:     linearize equation 17 into 6 and obtain the first order optimality condition: $\boldsymbol{P}\bar{\boldsymbol{W}} = \boldsymbol{p}$
4:     **if** $\boldsymbol{P}$ is non-singular **then**
5:        solve the linear system and get the solution on desired points
6:     **else**
7:        resample $\Omega_e$ until $\boldsymbol{P}$ is non-singular and get the solution
8:     **end if**
9: **end for**

---

It is worthwhile to note that we would better to sample in the $(0, \mathcal{O}(\frac{1}{|\lambda_1|})]$ as the term $e^{\lambda_1}$ decays very fast. Otherwise, $\boldsymbol{P}$ may be singular due to the round-off error. Furthermore, our model has the potential to be extended to solve stiff PDEs with both time and spatial variables by discretizing the spatial variable using FDM, which is left with a system of ODEs. Careful discussion is needed.

## 4 NUMERICAL EXPERIMENTS

In this section, six benchmark numerical experiments are carried out to test the performance of the model. All the numerical results are coded in Matlab 2024a and run on a desktop computer with an Intel core i5-12490F CPU, 32 GB DIMM RAM, and a NVIDIA GeFORCE RTX 2060 GPU. The computation time and the discrete $L^2$ error against the reference solution are two key variables to show the performance. And the reference solution is obtained either from the analytic solution or the solution calculated by the built-in stiff ODE solver ode15s in Matlab with $RelTol = 1 \times 10^{-14}$ for the cases where analytic solution is unknown. All the experiments are run for 10 times on the same device and the results are the average value. For every iteration, the weight matrix is trained by solving a linear system by the built-in function $pinv$, which is a fast and stable solver for computing the Moore-Penrose pseudoinverse of a matrix. The adjustable parameters $\omega_i$ and $\omega_e$ are chosen as 1 in all experiments.

### 4.1 STIFF SINGLE ODE

We first test our model on a stiff single ODE tested by reduced PINNNasiri & Dargazany (2022) which is:

$$\begin{cases} y' - \lambda y = e^{-t}, t \in (0,1], \\ y(0) = 2. \end{cases} \tag{20}$$

The exact solution is:

$$y_{exact}(t) = (2 + \frac{1}{1+\lambda})e^{\lambda t} - \frac{e^{-t}}{1+\lambda}. \tag{21}$$

Under our format, we denote $y(t)$ as $y_1(t)$ and add a new function $y_2(t)$. To make $e^{-t}/y_2(t)$ as simple as possible, we choose $y_2(t) = e^{-t}$. Then the system 21 can be transited into:

$$\begin{cases} \begin{pmatrix} y_1' \\ y_2' \end{pmatrix} = \begin{pmatrix} \lambda & 1 \\ 0 & -1 \end{pmatrix} \begin{pmatrix} y_1 \\ y_2 \end{pmatrix}, \\ y_1(0) = 2, y_2(0) = 1. \end{cases} \tag{22}$$

For this simple case, the two eigenvalues are $\lambda$ and $-1$ and the activation functions are 23 if $\lambda \neq -1$.

$$\sigma(t) = (e^{\lambda t}, e^{-t})^T. \tag{23}$$

The original problem is tested for $\lambda = -50$. For our model, we test on a much stiffer case where $\lambda = -1000$. The result is shown in figure 4.1.

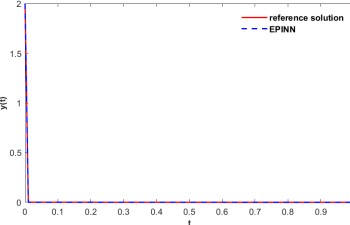

Figure 1: The result of equation 20 with 10 collation points chosen randomly in $(0, 0.1]$. The average computation time is almost $0s$ and the discrete $L^2$ error is $4.4106 \times 10^{-14}$ after running for 10 times.

### 4.2 LINEAR STIFF ODES WITH CONSTANT COEFFICIENTS

We then test our model on the following linear system of stiff ODEs with constant coefficients which is used to benchmark stiff ODE solvers by Stoer et al. (1980):

$$\begin{cases} y_1' = (\frac{\lambda_1 + \lambda_2}{2})y_1 + (\frac{\lambda_1 - \lambda_2}{2})y_2, \\ y_2' = (\frac{\lambda_1 - \lambda_2}{2})y_1 + (\frac{\lambda_1 + \lambda_2}{2})y_2, \\ y_1(0) = 2, y_2(0) = 0. \end{cases} \tag{24}$$

where $t \in [0, 1]$. The exact solutions is:

$$\begin{cases} y_1(t) = e^{\lambda_1 t} + e^{\lambda_2 t}, \\ y_2(t) = e^{\lambda_1 t} - e^{\lambda_2 t}. \end{cases} \tag{25}$$

The original paper sets $\lambda_1$ and $\lambda_2$ as -20 and -2, with stiffness ratio 10. We test our model with a stiffer case where $\lambda_1 = -1000, \lambda_2 = -1$ and the stiffness ratio is 1000. The result of $y_1$ and $y_2$ is shown in figure 4.2.

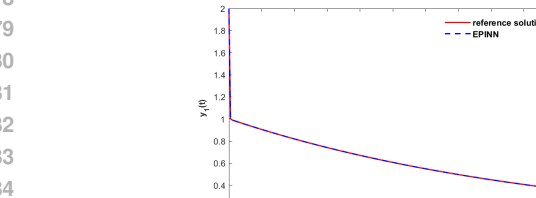 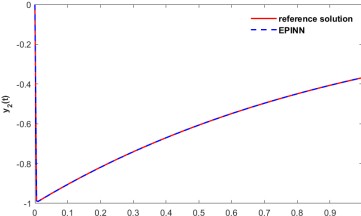

Figure 2: The result of equation 24 with 10 collation points chosen randomly in $(0, 0.1]$. After running for 10 times, the average computation time is almost $0s$ and the discrete $L^2$ errors for $y_1, y_2$ are $6.9799 \times 10^{-11}$ and $6.9803 \times 10^{-11}$, respectively.

### 4.3 LINEAR STIFF ODEs WITH POLYNOMIAL COEFFICIENTS

We test our model against the following Linear Stiff ODEs with polynomial coefficients:

$$\begin{cases} y' = \begin{pmatrix} -4.5t^2 - 4t - 199 & -1.5t^2 + 99 \\ -1.5t^2 + 2t - 198 & -4.5t^2 - 2t + 98 \end{pmatrix} y, \\ y(0) = (1, 1)^T. \end{cases} \tag{26}$$

where $t \in [0, 1]$. This is a stiff equation whose solution does not have explicit form and the reference solution is obtained from ode15s. As the coefficients is a polynomial matrix function, we can construct our network by calculating all the eigenvalues either manually or by the computer. After computation, we get our activation function as: $\sigma(t) = \sigma_1(t) \otimes \sigma_2(t) \otimes \sigma_3(t)$ with:

$$\sigma_1(t) = (e^{-t^3}; e^{-2t^3}), \sigma_2(t) = (e^{-t^2}; e^{-2t^2}), \sigma_3(t) = (e^{-t}; e^{-100t}). \tag{27}$$

The results of $y_1$ and $y_2$ are shown in figure 4.3.

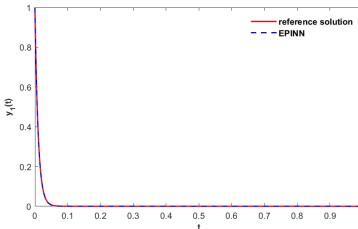 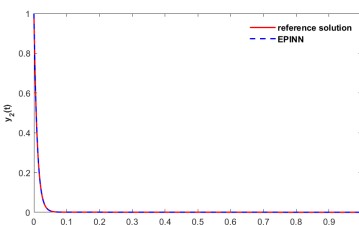

Figure 3: The result of equation 26. Sequential EPINN is employed in this case, where [0,1] is partitioned into 100 equidistant small intervals and 10 randomly chosen collation points is chosen as the sample set during each iteration. After running for 10 times, the average time is almost $0s$ and the $L^2$ error for $y_1$ and $y_2$ are $7.9811 \times 10^{-4}$ and $7.9859 \times 10^{-4}$, respectively.

### 4.4 PROTHERO-ROBINSON PROBLEM

The Prothero-Roberson benchmark problemProthero & Robinson (1974) is given by:

$$y' = \lambda(y - b(t)) + b'(t). \tag{28}$$

The analytical solution is $y(t) = b(t)$ and the problem becomes stiff for $|\lambda| \gg 1$. The solution to equation 28 is actually not stiff while the non-stiff ODE solver often fails to solve this problem for large parameter $\lambda$. We choose $b(t) = sin(t), y(0) = b(0) = 0, t \in [0, 2\pi]$ and $\lambda = -1 \times 10^5$ same as Fabiani et al. (2023). To transit system equation 28 in a homogeneous one, we introduce two new variables for this case. Let $y_1(t) = y(t), y_2(t) = sin(t), y_3(t) = cos(t)$, system equation 28 can be rewritten as:

$$\begin{pmatrix} y_1' \\ y_2' \\ y_3' \end{pmatrix} = \begin{pmatrix} \lambda & -\lambda & 1 \\ 0 & 0 & 1 \\ 0 & -1 & 0 \end{pmatrix} \begin{pmatrix} y_1 \\ y_2 \\ y_3 \end{pmatrix}. \tag{29}$$

The numerical result is shown in figure 4.4.

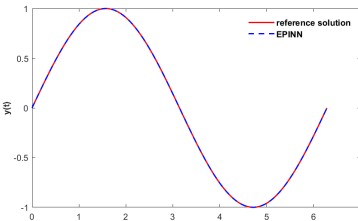

Figure 4: The result of equation 28 with 10 randomly chosen collation points in $(0, 1 \times 10^{-5}]$. After running for 10 times, the average time is almost $0s$ and the $L^2$ error is $2.1863 \times 10^{-6}$.

### 4.5 VAN DER POL'S EQUATION

Van Der Pol's equation is a benchmark stiff problem on studying nonlinear oscillations given by:

$$\begin{cases} y'' + \mu(y^2 - 1)y' + y = 0, \\ y(0) = 2, y'(0) = 0. \end{cases} \tag{30}$$

where $t \in [0, 3\mu]$. Sometimes one system would show stiffness when facing very small or large parameters. However, the system 30 would show great stiffness even when $\mu$ is not that large because of its non-linearity. By setting $y_1(t) = y(t), y_2(t) = y'(t)$, system equation 30 can be transited into the following first order ODEs system:

$$\begin{cases} \begin{pmatrix} y_1' \\ y_2' \end{pmatrix} = \begin{pmatrix} y_2 \\ -y_1 - \mu(y_1^2 - 1)y_2 \end{pmatrix}, \\ y_1(0) = 2, y_2(0) = 0. \end{cases} \tag{31}$$

The result of $y_1$ and its derivative $y_2$ is shown in figure 4.5.

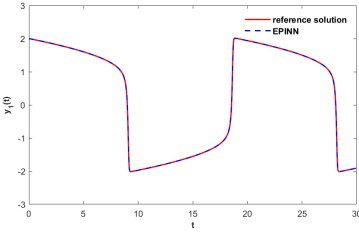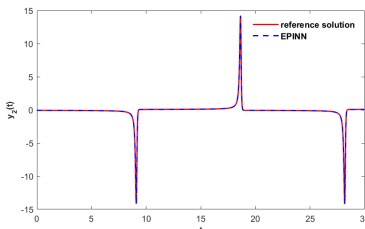

Figure 5: The result of equation 30. Sequential EPINN is implemented. For our case, $\mu$ is chosen as 10 and $[0, 3\mu]$ is partitioned into 3000 equidistant small intervals and 5 collation points are randomly chosen during each iteration. After running for 10 times, the average time is $0.0609s$ compared with $0.0063s$ consumed by ode15s. And the $L^2$ error for $y_1$ and $y_2$ are $0.0645$ and $0.7324$, respectively.

### 4.6 ROBER PROBLEM

Rober problem is a classical stiff equation which is to describe the reaction among three substances in chemical kinetics and is very popular in testing stiff ODE solvers. It can be written as the following equations:

$$\begin{cases} \begin{pmatrix} y_1' \\ y_2' \\ y_3' \end{pmatrix} = \begin{pmatrix} -0.04y_1 + 10^4 y_2 y_3 \\ 0.04y_1 - 3 \times 10^7 y_2^2 - 10^4 y_2 y_3 \\ 3 \times 10^7 y_2^2 \end{pmatrix}, \\ y_1(0) = 1, y_2(0) = 0, y_3(0) = 0. \end{cases} \tag{32}$$

where $t \in [0, 0.01]$. This example is quite different from what we have discussed because its Jacobi matrix is degenerate everywhere, which belongs to the category of differential algebraic equations(DAEs). It is easily observed that

$$y_1(t) + y_2(t) + y_3(t) \equiv 1, \tag{33}$$

According to this property, we can add a conservative loss term as:

$$L_c(W) = \frac{1}{2l} \sum_{i=1}^{l} \|y_1^*(t_i) + y_2^*(t_i) + y_3^*(t_i) - 1\|_2^2, \tag{34}$$

where $y_1^*, y_2^*, y_3^*$ are the estimated solutions by the network. The result of the concentration change of three substances is shown in figure 4.6.

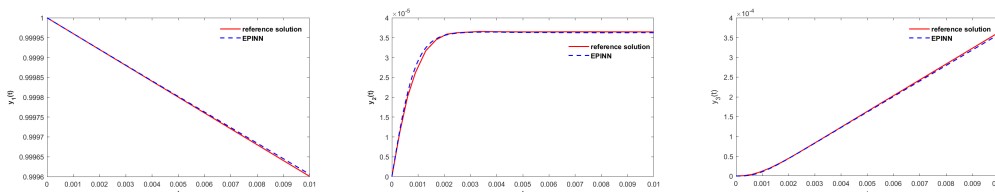

Figure 6: The result of equation 33. Sequential EPINN is also implemented in this case and we partition the time span into 200 equidistant sub-intervals. The sample set for governing equation loss term and mass conservative term is 2 and 10 randomly chosen collation points during each iteration, respectively. After running for 10 times, The average time is $0.0203s$ compared with almost $0s$ consumed by ode15s. And the $L^2$ error for $y_1$, $y_2$ and $y_3$ are $2.9559 \times 10^{-7}$, $6.3255 \times 10^{-8}$ and $2.8437 \times 10^{-7}$, respectively.

## 5 CONCLUSION

In this paper, we propose a brand new single layer PINN to solve the initial value problem of stiff ODEs by embedding the prior knowledge of the solution structure into the design of the network and the model shows great performance for both the linear and nonlinear stiff ODEs.

For linear systems, our model not only achieves satisfactory precision but also runs very fast. In our first four linear systems, the time consumed is all almost $0s$. For nonlinear ones, our scheme also holds a high calculation speed against regular multi-layer PINN models while maintaining great accuracy.

In the future, there is still an urge for designing higher-precision model to solve nonlinear stiff ODEs and extending the model to solve stiff PEDs with solid mathematical foundation and abundant numerical experiments as well. As for the issue this paper focus, one can try to develop the model with adaptive time step techniques to further enhance the efficiency and robustness.

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

## A APPENDIX

### A.1 SOLUTION TO LINEAR SYSTEMS WITH CONSTANT COEFFICIENTS

In this section, we prove the solution to equation 7 can be written as the combination $(1; \sigma_{\boldsymbol{A}}(t))$. First, the solution can be written in the exponential matrix function form:

$$\boldsymbol{y}(t) = e^{\boldsymbol{A}t}\boldsymbol{y}_0 + \int_0^t e^{\boldsymbol{A}(t-s)}\boldsymbol{b}\,ds. \tag{35}$$

where $e^{\boldsymbol{A}t} = \sum_{k=0}^{\infty} \frac{1}{k!}(\boldsymbol{A}t)^k$. Therefore, the key is to figure out the explicit expression of $e^{\boldsymbol{A}t}$.

By definition, when $\boldsymbol{A} = diag\{\boldsymbol{J}_1, \boldsymbol{J}_2, \ldots, \boldsymbol{J}_m\}$, $e^{\boldsymbol{A}} = diag\{e^{\boldsymbol{J}_1}, e^{\boldsymbol{J}_2}, \ldots, e^{\boldsymbol{J}_m}\}$ as different blocks would not bother each other when doing multiplication, which leads to:

$$
\begin{aligned}
&e^{\boldsymbol{A}t}, \\
=&e^{\boldsymbol{P}^{-1}\boldsymbol{J}\boldsymbol{P}t}, \\
=&\boldsymbol{I} + \boldsymbol{P}^{-1}\boldsymbol{J}\boldsymbol{P}t + \frac{1}{2!}(\boldsymbol{P}^{-1}\boldsymbol{J}\boldsymbol{P}t)^2 + \ldots, \\
=&\boldsymbol{P}^{-1}\boldsymbol{P} + \boldsymbol{P}^{-1}\boldsymbol{J}t\boldsymbol{P} + \frac{1}{2!}\boldsymbol{P}^{-1}\boldsymbol{J}^2 t^2 \boldsymbol{P} + \ldots, \\
=&\boldsymbol{P}^{-1}(\boldsymbol{I} + \boldsymbol{J}t + \frac{1}{2!}(\boldsymbol{J}t)^2 + \ldots)\boldsymbol{P}, \\
=&\boldsymbol{P}^{-1}e^{\boldsymbol{J}t}\boldsymbol{P}, \\
=&\boldsymbol{P}^{-1}
\begin{pmatrix}
e^{\boldsymbol{J}_1 t} & 0 & \ldots & 0 \\
0 & e^{\boldsymbol{J}_2 t} & \ldots & 0 \\
\vdots & \vdots & \ddots & \vdots \\
0 & 0 & \ldots & e^{\boldsymbol{J}_m t}
\end{pmatrix}
\boldsymbol{P}.
\end{aligned}
\tag{36}
$$

By the definition of $\boldsymbol{J}_i$ for each $i = 1, \ldots, m$, we have:

$$
\begin{aligned}
&e^{\boldsymbol{J}_i t}, \\
=&\boldsymbol{I} + \boldsymbol{J}_i t + \frac{1}{2!}(\boldsymbol{J}_i t)^2 + \cdots, \\
=&e^{\lambda_i t}
\begin{pmatrix}
1 & t & \ldots & \frac{t^{d_i-1}}{(d_i-1)!} \\
0 & 1 & \ldots & \frac{t^{d_i-2}}{(d_i-2)!} \\
\vdots & \vdots & \ddots & \vdots \\
0 & 0 & \ldots & 1
\end{pmatrix}.
\end{aligned}
\tag{37}
$$

On the other hand, notice the following commutativity holds:

$$e^{\boldsymbol{A}t}\boldsymbol{A}^{-1},$$

$$=(\sum_{k=0}^{\infty}\frac{1}{k!}(\boldsymbol{A}t)^k)\boldsymbol{A}^{-1},$$

$$=(\boldsymbol{I}+\boldsymbol{A}t+\frac{1}{2!}(\boldsymbol{A}t)^2+\dots)\boldsymbol{A}^{-1},$$

$$=(\boldsymbol{A}^{-1}+\frac{1}{2}\boldsymbol{A}t^2+\dots), \tag{38}$$

$$=\boldsymbol{A}^{-1}(\boldsymbol{I}+\boldsymbol{A}t+\frac{1}{2!}(\boldsymbol{A}t)^2+\dots),$$

$$=\boldsymbol{A}^{-1}e^{\boldsymbol{A}t}.$$

By using the fact that $(e^{-\boldsymbol{A}t})'=-\boldsymbol{A}e^{-\boldsymbol{A}t}$ and $\boldsymbol{A}$ is non-singular ($|\lambda_m|>0$), we get:

$$\boldsymbol{y}(t),$$

$$=e^{\boldsymbol{A}t}\boldsymbol{y}_0+e^{\boldsymbol{A}t}\int_0^t -\boldsymbol{A}^{-1}(e^{-\boldsymbol{A}s})'\boldsymbol{b}ds,$$

$$=e^{\boldsymbol{A}t}\boldsymbol{y}_0-e^{\boldsymbol{A}t}\boldsymbol{A}^{-1}(e^{-\boldsymbol{A}t}-\boldsymbol{I})\boldsymbol{b}, \tag{39}$$

$$=e^{\boldsymbol{A}t}(\boldsymbol{y}_0+\boldsymbol{A}^{-1}\boldsymbol{b})-\boldsymbol{A}^{-1}\boldsymbol{b},$$

$$=\boldsymbol{P}^{-1}e^{\boldsymbol{J}t}\boldsymbol{P}(\boldsymbol{y}_0+\boldsymbol{A}^{-1}\boldsymbol{b})-\boldsymbol{A}^{-1}\boldsymbol{b}.$$

Therefore we know $\boldsymbol{y}(t)$ is the linear combination of $(1;\sigma_1(t);\cdots;\sigma_m(t)) = (1;\sigma_{\boldsymbol{A}}(t))$. As a supplement, the basis function can be written out when $\boldsymbol{A}$ is singular, which needs more careful calculation.

### A.2 TECHNIQUE OF TRANSITING NON-HOMOGENEOUS SYSTEMS INTO HOMOGENEOUS ONES

In this section, we consider the following non-homogeneous system:

$$\begin{cases} \boldsymbol{y}'=\boldsymbol{A}(t)\boldsymbol{y}+\boldsymbol{b}(t), t\in(0,T], \\ \boldsymbol{y}(0)=\boldsymbol{y}_0. \end{cases} \tag{40}$$

One way to transit equation 40 into a homogeneous one is to add new variables. Generally speaking, one can consider adding a positive function $\boldsymbol{u}(t) \in \mathbb{R}^d$. Denoting $\boldsymbol{b}(t)/\boldsymbol{u}(t) = (b_1(t)/u_1(t); b_2(t)/u_2(t), \dots, b_d(t)/u_d(t))$, the criteria of choosing $\boldsymbol{u}(t)$ is to make $\boldsymbol{b}(t)/\boldsymbol{u}(t)$ as simple as possible. Two cases are shown in section 4.1 and section 4.4.

### A.3 SOLUTION TO LINEAR SYSTEMS WITH VARIABLE COEFFICIENTS

In this section, we discuss the structure of $\bar{\boldsymbol{y}}(t) = \Pi_{i=0}^k e^{\boldsymbol{A}_i t^{i+1}}\boldsymbol{y}_0$. Similar to the discussion in A.1, we know $e^{\boldsymbol{A}_i t^{i+1}}$ has the following structure by replacing $t$ with $t^{i+1}$:

$$e^{\boldsymbol{A}_i t^{i+1}} = \boldsymbol{P}_i^{-1}diag\{e^{\boldsymbol{J}_1^i},\dots,e^{\boldsymbol{J}_{m_i}^i}\}\boldsymbol{P}_i. \tag{41}$$

where $\boldsymbol{P}_i$ is a non-singular matrix and $e^{\boldsymbol{J}_j^i}$ is:

$$e^{\boldsymbol{J}_j^i} = e^{\lambda_j^i t^{i+1}} \begin{pmatrix} 1 & t^{i+1} & \cdots & \frac{(t^{i+1})^{d_j^i-1}}{(d_j^i-1)!} \\ 0 & 1 & \cdots & \frac{(t^{i+1})^{d_j^i-2}}{(d_j^i-2)!} \\ \vdots & \vdots & \ddots & \vdots \\ 0 & 0 & \cdots & 1 \end{pmatrix}. \tag{42}$$

where $\lambda_j^i$ is the eigenvalue of $\boldsymbol{A}_i$ with multiplicity $d_j^i$, $j = 1,\dots,m_i$. Therefore, $\bar{\boldsymbol{y}}(t)$ can be written as the linear combination of $\otimes_{i=0}^k \sigma_{\boldsymbol{A}_i}(t)$.

## A.4 Equivalent form of first order optimality condition

In this section, we discuss the first order optimality condition of equation 18. By denoting $(\boldsymbol{w}_1; \boldsymbol{w}_2; \ldots; \boldsymbol{w}_d) \in \mathbb{R}^{d^2}$ as $\bar{\boldsymbol{W}}$, we know the following property holds:

$$\boldsymbol{W}\sigma(0) = \begin{pmatrix} \sigma(0)^T & 0 & \cdots & 0 \\ 0 & \sigma(0)^T & \cdots & 0 \\ \vdots & \vdots & \ddots & \vdots \\ 0 & 0 & \cdots & \sigma(0)^T \end{pmatrix} \bar{\boldsymbol{W}}. \tag{43}$$

$$\boldsymbol{W}\sigma'(t_i) - \boldsymbol{A}(t_i)\boldsymbol{W}\sigma(t_i),$$

$$= \begin{pmatrix} \sigma'(t_i)^T - a_{11}(t_i)\sigma(t_i)^T & -a_{12}(t_i)\sigma(t_i)^T & \cdots & -a_{1d}(t_i)\sigma(t_i)^T \\ -a_{21}(t_i)\sigma(t_i)^T & \sigma'(t_i)^T - a_{22}(t_i)\sigma(t_i) & \cdots & -a_{2d}(t_i)\sigma(t_i)^T \\ \vdots & \vdots & \ddots & \vdots \\ -a_{d1}(t_i)\sigma(t_i)^T & -a_{d2}(t_i)\sigma(t_i)^T & \cdots & \sigma'(t_i)^T - a_{dd}(t_i)\sigma(t_i)^T \end{pmatrix} \bar{\boldsymbol{W}}. $$
$$\tag{44}$$

We denote the 43 as $\boldsymbol{P}_0\bar{\boldsymbol{W}}$ where $\boldsymbol{P}_0 \in \mathbb{R}^{d \times d^2}$ and 44 as $\boldsymbol{P}_i\bar{\boldsymbol{W}}$ where $\boldsymbol{P}_i \in \mathbb{R}^{d \times d^2}, i = 1, \ldots, n$. To solve the first order condition $\frac{\partial}{\partial \boldsymbol{W}} L(\boldsymbol{W}) = 0$ is equivalent to solve:

$$(\omega_i \boldsymbol{P}_0^T \boldsymbol{P}_0 + \frac{\omega_e}{n} \sum_{i=0}^{n} \boldsymbol{P}_i^T \boldsymbol{P}_i)\bar{\boldsymbol{W}} = \omega_i \boldsymbol{P}_0^T \boldsymbol{y}_0 + \frac{\omega_e}{n} \sum_{i=0}^{n} \boldsymbol{P}_i^T \boldsymbol{b}. \tag{45}$$

which is in the form of $\boldsymbol{P}\bar{\boldsymbol{W}} = \boldsymbol{p}$. If $\boldsymbol{P}$ is non-singular, the uniqueness and success of solving is both guaranteed.

