# OpenReview forum: "EPINN: Physics-Informed Neural Network with exponential activation functions for solving stiff ODEs"
_ICLR.cc/2025/Conference — ICLR 2025 Conference Withdrawn Submission_

### Official Review · Reviewer_KUXW · 2024-10-17

**Soundness:** 2
**Presentation:** 2
**Contribution:** 1
**Rating:** 1
**Confidence:** 3

**Summary:**

The paper proposes that using an exponential activation function for physics-informed neural networks (PINN) benefits optimization on stiff ordinary differential equations (ODEs). The modification of the network architecture is motivated by the solution theory of linear ODEs and experimentally tested on various stiff ODEs.

As I understand the paper, I see several issues:
1) The practical impact is limited as the focus is on linear systems. For linear ODEs, the solution theory is fairly complete. The authors’ method absorbs the time-dependency of the solution in the activations, or features, and this leads to a quadratic objective which they solve via inversion of the Hessian. But the whole procedure requires knowing the solution modes beforehand.
2) In section 4.1, the methodology is applied incorrectly. The system matrix, in the paper denoted by A, is 1x1. Yet, their basis function vector sigma is has two components, where the a second basis function was added that is read off the exact solution. Coincidentally, the inhomogeneous part is also exponential but for a generic inhomogeneous linear ODE, the method appears inapplicable.
3) In one experiment on a nonlinear systems, the time interval is decomposed into many (3000 in section 4.5) small portions, each with its own exponential PINN. In this setting an L2 error of 0.73 is extremely high. An absolute deviation of 0.01 on each of the 3000 subintervals of duration 0.01 would actually have a lower L2 error. I would assume the exponential PINN in each subinterval only learned to intersect the solution but not to approximate its form.
4) Generally, the experiments are minimal and its hard to draw conclusions from statements like ‘the average computation time is almost 0s’. There are no comparisons to other PINNs or other methods. This would be crucial in the absence of theoretical guarantees for nonlinear systems.
5) The related work section is about half a page. Given that PINNs are widely studied by now and improving PINNs through model architecture modifications as well, this seems rather short.

In my opinion, the changes needed to make this paper acceptable are extensive, so I recommend a reject. I would recommend to consider the following changes:
- Include other PINN methods and reproduce the issue of their slow convergence on stiff problems and show that the your method decreases the loss faster over a range of hyperparameter choices.
- Shift to more complex cases such as partial differential equations. Optimally, study a high-dimensional system or a system on a complex geometric shape, where PINNs have advantages over classical methods in terms of scalability and practicability.

Minor/typos:
- L83: ‘Without generality, we can always assume …’
- A space should be included before some citations.
- Eq. 5: what is ‘N’
- Some of the technical details in the experiments, should be moved to the appendix.

**Strengths:**

See above.

**Weaknesses:**

See above.

**Questions:**

What is the motivation to study PINNs on linear ODEs?

---

### Official Review · Reviewer_rnpv · 2024-10-23

**Soundness:** 2
**Presentation:** 3
**Contribution:** 1
**Rating:** 3
**Confidence:** 4

**Summary:**

Inspired by the solution structures of linear ODEs, the authors designed a single-layer neural network with exponential activation function and trained it in the PINN manner to solve the initial value problem of stiff ODEs. The model shows great performance for both linear and nonlinear stiff ODEs by achieving satisfactory precision while running very fast.

For increased accuracy and robustness, especially when dealing with nonlinear systems, the EPINN can be implemented sequentially. This involves partitioning the time interval into sub-intervals and solving each sub-problem separately, using the solution from the previous interval as the initial condition for the next.

**Strengths:**

The idea presentation comes with great clarity.

- The authors of the EPINN paper effectively explain their rationale behind choosing a single-layer architecture with exponential activation functions. They demonstrate this through a progression of examples, starting with simple linear ODEs (w/ and w/o constant coefficients) and advancing to more complex nonlinear ones, further motivating the sequential EPINN.

- The authors incorporated appropriate sanity checks to verify the method's effectiveness at each stage, with increasingly complex experimental objects, ensuring the validity of their results.

- The authors apply EPINN to challenging nonlinear examples, successfully showcasing its ability to handle such problems, the result presentation was easy to follow.

**Weaknesses:**

- Limited Novelty and Application scenario: The use of "activation functions inspired by the solution form of an ODE" in neural networks for has been explored in previous research. For instance, "Learning Compact Neural Networks Using Ordinary Differential Equations as Activation Functions", suggesting a bit of overlap with the core idea presented in EPINN. More importantly, the specific form of the activation function in EPINN is heavily reliant on the availability of prior knowledge about the system's dynamics - analytical derivations of linear ODEs w/ and w/o constant coefficients are wonderful materials for motivating the choice of the activation function, however, it also suggests the design depends highly on the prior knowledge and the specific forms of the system, which makes the method less generalizable.

- Architectural Simplicity and Its Implications to contribution of PINN: The single-layer architecture of EPINN, while computationally efficient, raises questions about its categorization within the broader field of deep learning. Optimizing a single-layer neural network often aligns more closely with traditional machine learning and optimization techniques. This is because the expressive power often raised from the complexity of multi-layered networks, might not be fully leveraged in this context, limiting its contribution to the advancement of the PINN literature. Could the authors comment on how EPINN's single-layer architecture relates to or advances the state-of-the-art in physics-informed neural networks, if I missed anything important?

- Comprehensive Experimental Comparison and Analysis: The experimental evaluation would benefit from a more comprehensive comparison against both classical numerical methods and alternative PINN approaches. While the comparison with ode15s is valuable, including results from other widely used stiff ODE solvers (e.g., ode23s, ode23t, ode23tb) would offer a more complete performance landscape. Additionally, benchmarking EPINN against traditional multi-layer PINNs is crucial to substantiate the claim of superior computational speed. The presentation should include a detailed analysis of the strengths and weaknesses of each method (EPINN, ode15s, other stiff ODE solvers, multi-layer PINNs). This analysis should extend beyond a simple list of pros and cons and provide insights into the factors driving their performance characteristics.

- Mathematical Rigor in Sequential EPINN: The paper acknowledges the local nature of certain approximations in EPINN and proposes the use of sequential EPINN to address this limitation. However, a more rigorous mathematical treatment of this sequential approach is necessary. This includes a clear explanation of the mechanism used to combine solutions from individual sub-problems, along with a discussion of potential error propagation across these stages. A rigorous error analysis is required - how to combine the approximation error bound of the single layer neural network with other approximation steps and derive a unified error bound? This analysis should leverage established error control theory for single-layer neural networks and provide theoretical guarantees for the accuracy of the sequential approach.

**Questions:**

It is claimed on page 10, line 522 that "our scheme holds a high calculation speed against regular multi-layer PINN models", however, I did not find any reported numbers for regular PINN ...? If it is missing, could the authors provide quantitative runtime comparisons between EPINN and standard PINN approaches for the benchmark problems presented?

---

### Official Review · Reviewer_Xfsg · 2024-11-01

**Soundness:** 1
**Presentation:** 2
**Contribution:** 1
**Rating:** 1
**Confidence:** 4

**Summary:**

The proposed method's reliance on fewer parameters compared to traditional PINNs is a significant advantage. The authors test EPINN on several benchmark problems, demonstrating the model's applicability across various scenarios. However, the paper lacks a detailed analysis of how EPINN achieves this balance compared to traditional methods. Given the existence of many numerical methods for solving stiff ODEs, it is unclear why EPINN is necessary for addressing such problems. The paper should include comparisons with other state-of-the-art methods specifically designed for stiff ODEs, such as specialized numerical solvers. Additionally, it does not provide sufficient metrics or comparisons regarding training time and convergence rates.

**Strengths:**

The proposed method's reliance on fewer parameters compared to traditional PINNs is a significant advantage. The authors test EPINN on several benchmark problems, demonstrating the model's applicability across various scenarios.

**Weaknesses:**

The paper lacks a detailed analysis of how EPINN achieves this balance compared to traditional methods. Given the existence of many numerical methods for solving stiff ODEs, it is unclear why EPINN is necessary for addressing such problems. The paper should include comparisons with other state-of-the-art methods specifically designed for stiff ODEs, such as specialized numerical solvers.  More specifically, it would be helpful to compare EPINN with methods like RADAU5, SDIRK, or TR-BDF2, which are known for their effectiveness with stiff ODEs.
Additionally, it does not provide sufficient metrics or comparisons regarding training time and convergence rates. for instance, computing time, the number of iterations, and error rates v.s. iterations.

**Questions:**

All the numerical examples presented are relatively simple ODEs that traditional methods can handle effectively. It is unclear why EPINN is necessary to solve these problems. What is the novelty of this approach in comparison to existing methods? Since neural networks are known to overcome the curse of dimensionality, I am interested in seeing how your method performs on high-dimensional ODEs or even PDEs.

---

### Official Review · Reviewer_6sss · 2024-11-02

**Soundness:** 2
**Presentation:** 2
**Contribution:** 2
**Rating:** 5
**Confidence:** 4

**Summary:**

PINNs are commonly used for solving differential equations, but they may trade precision for speed. Stiff ODEs which may describe multiscale physical systems require both numerically stable and efficient solvers, but there are currently few PINN models that might be applicable to solving such problems. The present paper proposes a novel shallow PINN model with exponential activation functions (EPINN) where the prior knowledge of the solution of stiff linear ODEs is incorporated in the model.

**Strengths:**

The paper demonstrates that the proposed model can solve exactly linear ODEs and provide reasonable approximations to non-linear ODEs. Furthermore, given the choice of a shallow architecture and exponential activation functions the model training can be more efficient due to the smaller number of parameters and the convex shape of the loss function.

**Weaknesses:**

The proposed model heavily relies on the identifiability of the solution structure of the solved differential equation.

Not enough information on the particular number of neurons of the NN, optimiser, training epochs, etc. is provided that would allow for reproducibility of results.

**Questions:**

The authors suggest that EPINN can be applied to solve stiff PDEs. Would any linearisation of the PDE be required beforehand? What would be your general approach?

Which of the features of the proposed model actually are most strongly related to its ability to solve stiff DEs in particular?

As a minor point, some additional proofreading can help removing some typos.

---

### Note · Authors · 2024-11-17

I have read and agree with the venue's withdrawal policy on behalf of myself and my co-authors.